# The Dual Role of Oat Bran Water Extract in Bone Homeostasis Through the Regulation of Osteoclastogenesis and Osteoblast Differentiation

**DOI:** 10.3390/molecules23123119

**Published:** 2018-11-28

**Authors:** Shin-Hye Kim, Kwang-Jin Kim, Hyeon Jung Kang, Young-Jin Son, Sik-Won Choi, Mi-Ja Lee

**Affiliations:** 1Division of Crop Foundation, National Institute of Crop Science (NICS), Rural Development Administration (RDA), Wanju 55365, Korea; black7a@naver.com (S.-H.K.); happykorean@korea.kr (H.J.K.); 2Department of Biological Sciences, College of Natural Science, Chonbuk National University, Jeonju 54896, Korea; 3Department of Pharmacy, Sunchon National University, Suncheon, Jeonnam 57922, Korea; mastiffk@naver.com (K.-J.K.); sony@sunchon.ac.kr (Y.-J.S.); 4Forest Biomaterials Research Center, National Institute of Forest Science (NIFS), Jinju, Gyeongnam 52817, Korea

**Keywords:** bone, osteoclasts, osteoblasts, oat bran, osteoporosis

## Abstract

The number of patients with bone metabolic disorders including osteoporosis is increasing worldwide. These disorders often facilitate bone fractures, which seriously impact the patient’s quality of life and could lead to further health complications. Bone homeostasis is tightly regulated to balance bone resorption and formation. However, many anti-osteoporotic agents are broadly categorized as either bone forming or anti-resorptive, and their therapeutic use is often limited due to unwanted side effects. Therefore, safe and effective therapeutic agents are needed for osteoporosis. This study aims to clarify the bone protecting effects of oat bran water extract (OBWE) and its mode of action. OBWE inhibited RANKL (receptor activator of nuclear factor-κB ligand)-induced osteoclast differentiation by blocking c-Fos/NFATc1 through the alteration of I-κB. Furthermore, we found that OBWE enhanced BMP-2-stimulated osteoblast differentiation by the induction of Runx2 via Smad signaling molecules. In addition, the anti-osteoporotic activity of OBWE was also evaluated using an in vivo model. OBWE significantly restored ovariectomy-induced bone loss. These in vitro and in vivo results showed that OBWE has the potential to prevent and treat bone metabolic disorders including osteoporosis.

## 1. Introduction

Fractures caused by bone metabolic disorders including osteoporosis are recognized as a serious public health issue worldwide [1]. Osteoporotic fragility fractures can cause considerable pain and severe disability, which can reduce the quality of life. Therefore, protection against bone fragility is an important means of improving the quality of life.

The maintenance of bone homeostasis depends on both the number and the activity of bone-resorbing osteoclastic and bone-forming osteoblastogenic cells [2]. The imbalance between bone formation and resorption is a major cause of pathological bone disorders resulting in osteoporosis, rheumatoid arthritis, Paget’s disease, and periodontal disease [3,4,5]. Numerous pharmaceutical agents, which are broadly categorized as either anabolic or anti-resorptive, have been developed to treat bone metabolic diseases including osteoporosis. Although several effective agents are currently available for the treatment of osteoporosis, their use is often limited due to safety issues associated with side effects and long-term use. Accordingly, new therapies that have both bone forming and anti-resorptive effects with satisfactory safety assessments would be valuable to mitigate osteoporotic bone loss.

Osteoclasts are large multinucleated cells that are differentiated from hematopoietic stem cells and are responsible for bone resorption. Mature multinucleated osteoclast cells (MNCs) are differentiated by the fusion of osteoclast precursor cells (monocytes and macrophages) in response to two essential cytokines, macrophage colony-stimulating factor (M-CSF, also known as CSF1) and receptor activator of nuclear factor-κB ligand (RANKL, also known as TNFSF11). The binding of RANKL to RANK generates the activation of several signaling molecules including nuclear factor-κB (NF-κB), Akt, and MAP kinases, which are required for the expression of genes necessary for osteoclast differentiation [6]. In response to RANKL, these signaling molecules contribute to the regulation of the AP-1 transcription factor member, c-Fos, and the nuclear factor of activated T cells, cytoplasmic 1 (NFATc1), which are known to be key regulators for osteoclast differentiation, fusion, and maturation [7,8,9]. NFATc1, which requires c-Fos for its induction, regulates the process of osteogenesis by controlling osteoclast-related genes including tartrate-resistant acid phosphatase (TRAP, also known as ACP5), dendritic cell-specific transmembrane protein (DC-STAMP), and cathepsin K [10,11]. Apparently, transcription factors such as c-Fos and NFATc1 play a critical role in the regulation of molecules for osteoclast differentiation. Therefore, the pharmacological inhibition of osteoclast differentiation-mediated transcription factors is effective to overcome bone resorbing-mediated disorders [3,12].

Osteoblasts are pivotal cells for new bone formation and are differentiated from mesenchymal stem cells. Osteoblast differentiation is closely modulated by a variety of factors, including hormones, cytokines, transcription factors, and signaling pathways, which result in mineralization and osteoblastic differentiation for bone formation [13,14]. Bone morphogenetic proteins (BMPs), which belong to the transforming growth factor-β superfamily, are potent cytokines that stimulate osteoblast differentiation, bone formation, and regeneration [15,16]. BMP-2, a BMP family member, is a primary regulator of osteogenesis and activates intracellular signaling molecules through Smad 1/5/8 phosphorylation [17]. These signaling pathways collaborate with the expression of Runt-related transcription factor 2 (Runx2), which is one of the most important transcription factors [18]. The master regulator of osteoblast commitment, Runx2, regulates the modulation of osteogenesis-specific downstream effectors, including alkaline phosphatase (ALP), bone matrix protein encoding genes secreted phosphoprotein 1 (OPN; osteopontin, also known as Spp1), and bone gamma-carboxyglutamate protein (OCL; osteocalcin, also known as Bglap) [19,20]. Consequently, Runx2 has significant potential to be a target for new therapies that prevent bone loss by mitigating bone formation.

Many plants have been reported to exhibit a wide range of biological activity and are valuable sources of medicinal compounds [21,22]. Crops have been traditionally ingested for nutirion and health benefits, but their production byproducts are generally discarded or used as feedstuffs. However, byproducts from crops have been shown to contain various bioactive components including vitamins, minerals, and phytochemicals. In particular, oat bran is well known as a functional food that contains β-glucan, the heart-healthy soluble fiber, phenolic acids, flavonoids, steroidal saponins, and tocols [23,24,25,26]. In addition, it has been reported to exhibited antidiabetic, anticancer, antioxidative, anti-obesity, and lowing cholesterol activity [27,28,29,30,31]. However, the effect of oat bran extract on bone homeostasis has not been previously studied. Therefore, we investigated the effect of oat bran water extract (OBWE) on RANKL-induced osteoclast differentiation and BMP-2-dependent osteoblast differentiation. The studied mode of action revealed that the oat bran extract potentiated the bone homeostasis mechanism.

## 2. Results and Discussion

### 2.1. OBWE Inhibits RANKL-Induced Osteoclast Differentiation

Osteoclasts are present only in bone, and they play a critical role in bone resorption. The differentiation and activation of osteoclasts are mainly regulated by RANKL [32]. The inhibition of osteoclastogenesis has been presumed to be an effective treatment for pathological bone loss [33]. Therefore, we examined the anti-osteoclastogenic effect of OBWE on RANKL-induced osteoclast differentiation. As shown in Figure 1A, OBWE strongly inhibited the formation of TRAP-positive multinucleated osteoclast cells (TRAP + MNCs) in a dose-dependent manner. The inhibitory effect was confirmed by counting the number of TRAP + MNCs (Figure 1B; left panel) and by measuring TRAP activity (Figure 1B; right panel). To exclude the possibility that the inhibitory effect of OBWE on osteoclastogenesis was due to its cytotoxicity per se, its effect on the survival of bone marrow macrophage cells (BMMs) was further evaluated using a CCK-8 assay. As shown in Figure 1C, OBWE did not exhibit any cytotoxicity at the doses used to assess its inhibitory effects. These results suggest that OBWE meaningfully attenuated RANKL-induced osteoclastogenesis without apparent cytotoxicity.

### 2.2. OBWE Inhibits RANKL-Related Expression of c-Fos and NFATc1 through the Modulation of NF-κB/I-κB Signaling Molecules

To better understand how OBWE impaired osteoclast differentiation, we evaluated the osteoclastogenesis-mediated molecules including RANKL-induced transcriptional factors and early signaling pathways. RANKL stimulation in BMMs leads to the induction of major transcription factors required for osteoclast differentiation. In particular, c-Fos is expressed in the early stages of osteoclast formation and controls the induction of NFATc1, a master regulator of osteoclast differentiation [34,35]. Furthermore, c-Fos-deficient mice develop osteopetrosis due to the lack of osteoclast lineage commitment, and the expression of NFATc1 by RANKL in c-Fos-knockout mice is also abolished [11,36]. Therefore, the coaction of c-Fos and NFATc1 is indispensable to modulate RANKL-induced osteoclast differentiation. In this study, OBWE treatment attenuated the RANKL-induced mRNA expression levels of c-Fos and NFATc1, and their target molecules, such as TRAP, OSCAR, DC-STAMP, and Cathepsin K (Figure 2A). Western blot analysis further revealed that the RANKL-induced protein expression of c-Fos and NFATc1 was also inhibited by OBWE treatment (Figure 2B). These results suggest that the inhibitory effect of OBWE on osteoclast differentiation is involved in the modulation of c-Fos/NFATc1, which functions as a transcriptional marker.

To gain insight into the mechanism by which OBWE inhibits osteoclast differentiation by attenuating c-Fos/NFATc1 expression, we investigated whether OBWE could affect the activation of the RANKL-mediated several signaling molecules that are associated with the regulation of master transcription factors. RANKL signaling during osteoclast differentiation activates various signaling pathways including NF-κB, PI3K/AKT, and MAP kinases [32,37]. In particular, the expression of c-Fos and NFATc1 requires the assembly of NF-κB signaling pathways [38]. NF-κB p50/p52 double deficiency mice show acute osteopetrosis and weaknesses in osteoclastogenesis because c-Fos and NFATc1 are not expressed by RANKL stimulation [39]. The function of NF-κB proteins is regulated by I-κB signaling molecules. I-κB inhibits NF-κB by preventing the nuclear localization signals of NF-κB molecules, thus segregating and maintaining them in an inactive state in the cytoplasm [40]. As shown in Figure 2C, RANKL stimulated the degradation of I-κB and the phosphorylation of RAC-Alpha Serine/Threonine-Protein Kinase (AKT), but the treatment of OBWE only prevented the RANKL-induced degradation of I-κB. These results suggest that the inhibition of I-κB degradation could be attributed to the anti-osteoclastogenic action of OBWE.

### 2.3. OBWE Enhances BMP-2-Mediated Osteoblast Differentiation in C2C12 Cells

Our results demonstrated that OBWE inhibits RANKL-induced osteoclast differentiation in BMMs, however, an appropriate balance of osteoclast and osteoblast is important for the effective treatment of bone metabolic disorders. Therefore, we examined whether OBWE could induce the bipotential of mesenchymal C2C12 cells to commit to osteoblast differentiation. BMP-2 enhances osteoblast differentiation by the induction of ALP activity and expression in C2C12 cells [41]. Treatment with OBWE dose-dependently enhanced BMP-2-mediated ALP expression, which was deduced by ALP staining (Figure 3A). Consistent with this result, OBWE significantly enhanced BMP-2-induced ALP activity in a dose-dependent manner (Figure 3B). However, no cytotoxicity of OBWE was observed at the doses used to assess its effects (Figure 3C). These results indicate that OBWE may possess osteogenic activity without apparent cytotoxicity.

### 2.4. OBWE Contributes to the BMP-2-Stimulated Expression of Runx2 through the Activation of Smad Signaling Pathways

To clarify the mode of the osteogenic action of OBWE, we further investigated the osteogenic mechanism in BMP-2-stimulated osteoblast differentiation. BMP-2 stimulation has shown to regulate osteogenic transcription factors such as Runx2, which play a central role in osteoblast differentiation [42]. The vital role of Runx2 in bone formation has been reported in heterozygous knockout mice which have osteopenia due to reduced osteoblastic function [43]. In addition, the activation of Runx2 leads to the regulation of osteoblast-specific molecules including ALP, OPN, and OCL, which are required for osteoblast differentiation and bone formation [41,44]. Therefore, we evaluated the effect of OBWE on the BMP-2-stimulated expression level of Runx2 and its downstream osteoblast-sepecific molecules. As shown in Figure 4A, *Runx2* was synergistically enhanced with the treatment of OBWE, and the transcription factor-regulated molecules *ALP*, *OPN*, and *OCL* were also stimulated on the indicated days. Consistent with previous data, BMP-2 induction of the translational expression of both Runx2 and ALP was synergistically enhanced by the addition of OBWE (Figure 4B). These results indicate that the anabolic activity of OBWE has the potential to stimulate the expression of Runx2, which is required for osteogenesis.

To investigate the underlying mechanism of the osteogenic activity of OBWE, we examined whether OBWE affects the activation of the BMP-mediated signaling molecules, which are major transcription factors required for bone formation, associated with the regulation of Runx2. In osteoblast differentiation, BMP-2 transmits an osteogenic intracellular signaling axis through a Smad-reliant pathway [45,46]. The critical process in the activation of BMP-2 signaling is the phosphorylation and translocation of Smad1, Smad5, and Smad8 (Smad 1/5/8) [47]. In addition, BMP-2-stimulated Smad proteins regulate the transcription of Runx2, OCL, and OPN, which play a central role in bone formation [42,48,49]. In this study, OBWE synergistically induced the BMP-stimulated phosphorylation of Smad (Figure 5). These results suggest that the anabolic effect of OBWE on BMP-2-dependent osteogenesis could result from its ability to enhance the Smad-Runx2 signaling axis for bone formation.

### 2.5. OBWE Prevents Ovariectomy-Induced Bone Loss In Vivo

The in vitro anti-osteoporotic effect of OBWE prompted us to explore its putative in vivo effects using an osteoporosis model with ovariectomy (OVX). OVX-induced bone loss in mice is widely used as a model of postmenopausal osteoporosis and is validated as a clinically relevant model of this condition in humans. For the OVX-stimulated bone loss in the animal model used in this study, micro-CT imaging indicated that the trabecular bone in the distal metaphyseal region of the femur was decreased by OVX, and an orally administration of OBWE dramatically prevented OVX-induced trabecular bone loss (Figure 6A). In addition, the OVX-induced changes of bone mineral density (BMD), percent bone volume ratio (BV/TV), and trabecular separation (Tb. Sp) were substantially protected by OBWE administration (Figure 6B). These results demonstrate that it is likely that OBWE can prevent and improve postmenopausal osteoporosis.

## 3. Materials and Methods

### 3.1. Preparation of the Oat Bran Water Extract

The oat bran used in this study was grounded in a laboratory test mill (Brabender Technologie, Duisburg, Germany). The flour (100 g) was defatted three times with hexane (1 L) for 24 h at room temperature. After filtration with filter paper (Whatman No. 2), the residual oat bran was extracted three times with prethanol (1 L) and filtrated by means of a Buechner funnel lined with filter paper (Carl Roth, Karlsruhe, Germany, 111A, Ø100 mm). The filtrates were combined and concentrated in a rotary evaporator. The residuals from the ethanol extraction were extracted twice with water (1 L) for 24 h at room temperature and combined and dried with a freeze dryer.

### 3.2. Reagents and Antibodies

Mouse-soluble RANKL, M-CSF, recombinant human bone morphogenetic protein-2 (rhBMP-2), and ALP antibodies were purchased from R & D Systems (Minneapolis, MN, USA). Penicillin, streptomycin, cell culture medium and foetal bovine serum (FBS) were purchased from Invitrogen Life Technologies (Carlsbad, CA, USA). Antibodies against c-Fos, NFATc1, actin, IκB, Smad, and the secondary antibody conjugated to horseradish peroxidase (HRP) were purchased from Santa Cruz Biotechnology (Dallas, TX, USA). All other antibodies were obtained from Cell Signaling Technology (Beverly, MA, USA).

### 3.3. Preparations of Osteoclast Precursor Cells

The osteoclast study was conducted in strict accordance with the recommendations of the Standard Protocol for Animal Study of Gangnam Severance Hospital Biomedical Center (Permit No. 2016-0238). The protocol (ID No. 0238) was approved by the Institutional Animal Care and Use Committee (IACUC) of Yonsei University College of Medicine. Every effort was made to minimise the number of animals used in the study and minimise their suffering and stress/discomfort.

All experiments were carried out as described in a previous study, with modifications [50]. Five-week-old male imprinting control region (ICR) mice (Damul Science Co., Deajeon, Korea) were maintained in a room illuminated daily from 07:00 to 19:00 (12:12 h light to dark cycle), with controlled temperature (23 ± 1 °C) and ventilation (10–12 times per h). Humidity was maintained at 55 ± 5% and the animals had free access to a standard animal diet and tap water. Bone marrow cells were obtained from the five-week-old male ICR mice by flushing their femurs and tibias with alpha minimum essential medium (α-MEM) supplemented with antibiotics (100 units/mL penicillin, 100 μg/mL streptomycin). The bone marrow cells were cultured on culture dishes for 1 d in α-MEM containing 10% FBS and M-CSF (10 ng/mL). The non-adherent bone marrow cells were plated onto Petri dishes and cultured for 3 d in the presence of M-CSF (30 ng/mL). After the non-adherent cells were washed out, the adherent cells were used as bone marrow-derived macrophages (BMMs).

### 3.4. Osteoclast Cell Culture and Osteoclast Differentiation

The BMMs were maintained in α-MEM supplemented with 10% FBS, 100 units/mL penicillin, and 100 μg/mL streptomycin. The medium was changed every 3 d in a humidified atmosphere of 5% CO_2_ at 37 °C. To differentiate the osteoclasts from the BMMs, the BMMs (1 × 10^4^ cells/well in a 96-well plate or 3 × 10^5^ cells/well in a 6-well plate) were cultured with M-CSF (30 ng/mL) and RANKL (10 ng/mL) for 4 d, and then the multinucleated osteoclasts were observed.

### 3.5. TRAP Staining and Activity Assay

The mature osteoclasts were visualized using TRAP staining, a biomarker of osteoclast differentiation. Briefly, the multinucleated osteoclasts were fixed with 3.7% formalin for 10 min, permeabilized with 0.1% Triton X-100 for 10 min, and then stained with TRAP solution (Sigma-Aldrich, Saint Louis, MO, USA). The TRAP-positive multinucleated osteoclasts (MNC; nuclear ≥ 3) were counted. To measure TRAP activity, the multinucleated osteoclasts were fixed in 3.7% formalin for 5 min, permeabilized with 0.1% Triton X-100 for 10 min, and then treated with TRAP buffer (100 mM sodium citrate, pH 5.0, 50 mM sodium tartrate) containing 3 mM *p*-nitrophenyl phosphate (Sigma-Aldrich) at 37 °C for 5 min. The reaction mixtures in the wells were transferred to new plates containing an equal volume of 0.1 N NaOH, and the optical density values were determined at 405 nm.

### 3.6. Cell Viability Assay

The BMMs and C2C12 cells were plated on 96-well plates (three replicate plates) at a density of 1 × 10^4^ cells/well (BMMs) or 2.5 × 10^3^ cells/well (C2C12 cells). After treatment with the indicated concentrations of OBWE, the cells were incubated for 3 d, and cell viability was measured using the Cell Counting Kit 8 (CCK-8) according to the manufacturer’s protocol. The CCK-8 assay kit was purchased from Dojindo Molecular Technologies (Rockville, MD, USA).

### 3.7. RNA Isolation and Real-Time Polymerase Chain Reaction Analysis

Real-time polymerase chain reaction (PCR) was performed as previously described [50]. The primers were chosen using the online Primer3 design program [51]. The primer sets used in this study are shown in Table 1. Briefly, total RNA was isolated with TRIzol reagent, and the first-strand of the cDNA was synthesized with the RevertAid First Strand cDNA Synthesis Kit (Thermo Scientific, Waltham, MA, USA) according to the manufacturer’s recommended protocol. SYBR green-based quantitative PCR (qPCR) was performed using the Bio-Rad CFX96 Real-Time PCR Detection System (Hercules, CA, USA) and Topreal qPCR 2 × PreMIX (Enzynomics, Daejeon, Korea). All reactions were run in triplicates, and the data were analyzed using the 2^−ΔΔCT^ method [52]. Hypoxanthine phosphoribosyltransferase 1 (HPRT1) and glyceraldehyde 3-phosphate dehydrogenase (GAPDH) were used as the internal standard genes. The statistical significance was determined using a Student’s *t*-test with HPRT1/GAPDH-normalized 2^−ΔΔCT^ values; the differences were considered significant at *p* < 0.05.

### 3.8. Western Blot Analysis

The Western blot analysis was performed as previously described [53]. Briefly, the cultured cells were washed with ice-cold phosphate-buffered saline (PBS) and lysed in lysis buffer (50 mM Tris-HCl, 150 mM NaCl, 5 mM EDTA, 1% Triton X-100, 1 mM sodium fluoride, 1 mM sodium vanadate, and 1% deoxycholate) supplemented with protease inhibitors. After centrifugation at 15,000× *g* for 15 min, the protein quantification of the supernatant was determined using the detergent compatible (DC) protein assay kit (Bio-Rad). The quantified proteins were denatured, separated on sodium dodecyl sulfate-polyacrylamide gel electrophoresis (SDS-PAGE) gels, and transferred onto a polyvinylidene difluoride (PVDF) membrane (Merck Millipore, Darmstadt, Germany). After incubation with an antibody, the membranes were developed using SuperSignal West Femto Maximum Sensitivity Substrate (Thermo Scientific) and visualized with the LAS-4000 luminescent image analyzer (GE Healthcare Life Sciences, Little Chalfont, UK). Actin and GAPDH were used as a loading control.

### 3.9. Osteoblast Differentiation

All osteoblast-mediated experiments were executed as previously described with modifications [54]. Mouse mesenchymal precursor C2C12 cells were from the American Type Culture Collection (Manassas, VA, USA). C2C12 cells were maintained in alpha minimum essential medium (α-MEM) with 10% FBS, 100 units/mL penicillin, and 100 μg/mL streptomycin. Cells were seeded in 96-well plates at 2.5 × 10^3^ cells/well or in 6-well plates at 2.5 × 10^5^ cells/well. After 1 d, cells were differentiated by replacing the medium with α-MEM containing 5% FBS and rhBMP-2 (100 ng/mL) with OBWE at the indicated dose. Osteoblastic bone formation was observed by ALP staining.

### 3.10. Alkaline Phosphatase Staining and Activity Assays

ALP is an early biomarker of osteoblast differentiation. After differentiation for 3 d, cells were washed twice with PBS, fixed with 10% formalin in PBS for 5 min, rinsed with deionized water, and stained using an ALP Kit (Sigma-Aldrich). To measure ALP activity, differentiated cells were washed twice with PBS, fixed with 10% formalin in PBS for 5 min, rinsed with PBS, and measured using a one-step PNPP substrate solution (Thermo Scientific, MA, USA) according to the manufacturer’s protocol.

### 3.11. Ovariectomy-Induced Bone Erosion

All procedures involving mice were conducted in strict accordance with Sunchon National University Institutional Animal Care and Use Committee (SCNU IACUC) guidelines for the care and use of laboratory animals (Permit Number: SCNU IACUC-2016-07). Five-week-old female ICR mice (n = 40: Damool Science, Korea) were divided into five groups of eight mice and received an ovariectomy (n = 32) or sham surgery (n = 8) by the dorsal approach under general anesthesia. The OVX + OBWE group was orally administered OBWE (1, 10, 100 mg/kg of body weight) and the control group was orally administered distilled water daily for 6 weeks from 1 d after surgery. The femurs were obtained from sacrificed mice by cervical dislocation and were fixed in 3.5% formaldehyde for 1 d. The fixed femurs were scanned and analyzed with the SkyScan 1272 micro-CT imaging system provided by Bruker (Billerica, MA, USA).

### 3.12. Statistical Analysis

All quantitative values are presented as mean ± standard deviation. Each experiment with three replicates for each experimental variable was performed three to five times, and Figure 1, Figure 2, Figure 3, Figure 4, Figure 5 and Figure 6 show the results from one representative experiment. Statistical differences were analyzed using the Student’s *t*-test, and a value of *p* < 0.05 was considered significant.

## 4. Conclusions

In summary, this is the first study to demonstrate that OBWE has a dual function in the inhibition of osteoclastogenesis and the induction of osteogenesis, although additional experiments are needed to substantiate the identification of the pharmaceutical compounds in OBWE that produce its anti-osteoporotic activity. In osteoclast differentiation, OBWE inhibits RANKL-mediated osteoclastogenic activity by preventing I-κB degradation and c-Fos/NFATc1 expression. The inhibition of osteoclastogenic-specific transcription factors by OBWE leads to the downregulation of biomarkers required for osteoclastogenic activity including TRAP, OSCAR, DC-STAMP, and cathepsin K. In osteoblast differentiation, the bone-forming effect of OBWE was associated with the induction of the Smad-Runx2 signaling molecules required for bone formation. Consequently, the induction of Runx2 could lead to the enhanced expression of the genes required for bone anabolic activity, such as ALP, OPN, and OCL. Furthermore, OBWE showed in vivo bone protection effects in an OVX-induced bone-loss model. The results presented here suggest that the apparent anti-osteoporotic property of OBWE could be a promising therapeutic substance for bone metabolic disorders including osteoporosis.

## Figures and Tables

**Figure 1 molecules-23-03119-f001:**
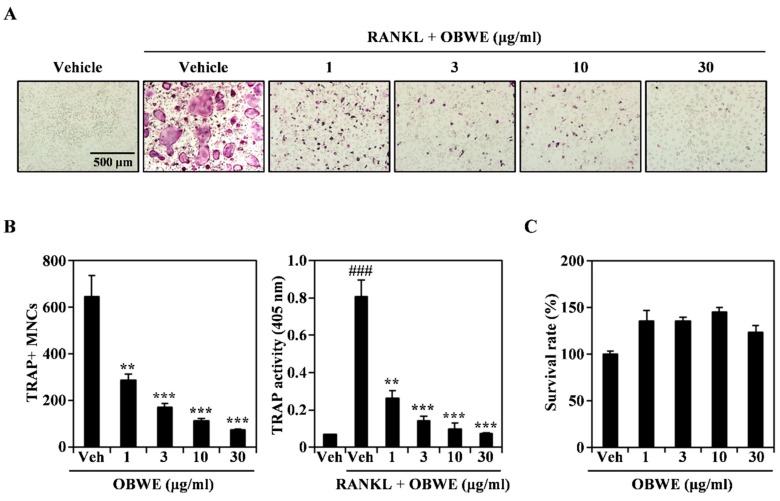
Oat bran water extract (OBWE) impairs RANKL-mediated osteoclast differentiation. (**A**) The BMMs were cultured for 4 d in the presence of RANKL (10 ng/mL) and M-CSF (30 ng/mL) with either the vehicle (water) or the indicated concentration of OBWE. Multinucleated osteoclasts were visualized using TRAP staining. (**B**) TRAP + MNCs were counted (left panel) and TRAP activity was measured (right panel). ^###^
*p* < 0.001 (versus the control); ** *p* < 0.01; *** *p* < 0.001 (versus the RANKL-treated group). (**C**) The effect of BSE on the viability of BMMs was evaluated using the CCK-8 assay. Data are expressed as mean ± SD and are representative of at least three experiments.

**Figure 2 molecules-23-03119-f002:**
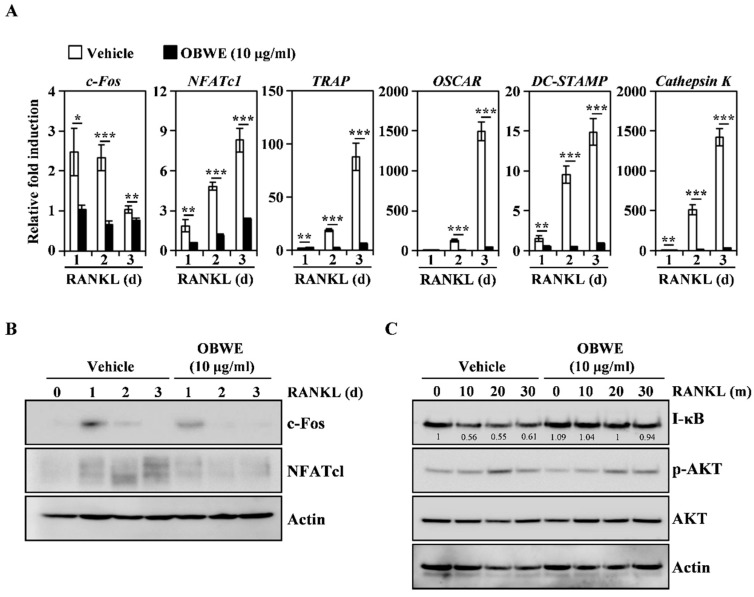
OBWE inhibits the RANKL-induced expression level of c-Fos/NFATc1 through the alteration of I-κB. (**A**) The BMMs were stimulated with RANKL (10 ng/mL) and M-CSF (30 ng/mL) in the presence or absence of OBWE (10 µg/mL) for the indicated times. Total RNA was then isolated using TRIzol reagent and the mRNA expression levels were evaluated using real-time PCR. *GAPDH* was used as the internal control. * *p* < 0.05; ** *p* < 0.01; *** *p* < 0.001 (versus the vehicle control). (**B**) The effect of OBWE on the protein expression level of RANKL-induced transcription factors was evaluated using Western blot analysis. Actin was used as the internal control. (**C**) Serum-starved (for 1 h) BMMs were pretreated with or without OBWE (10 µg/mL) for 1 h prior to RANKL stimulation (10 ng/mL) at the indicated time periods. Then, the expression levels of the signaling molecules were evaluated using western blot analysis. The indicated densitometric values were obtained using Multi Gauge version 3 software. One representative result obtained from three independent experiments yielding similar results is shown.

**Figure 3 molecules-23-03119-f003:**
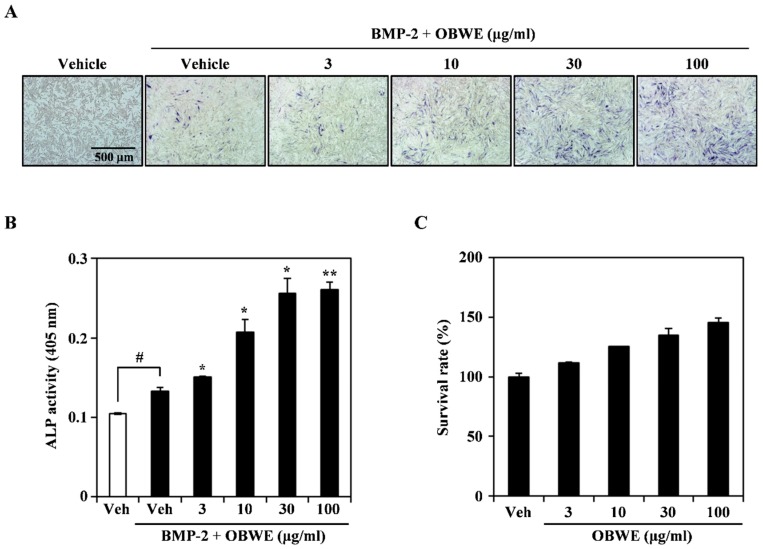
OBWE promotes BMP-2-induced osteoblast differentiation. (**A**) C2C12 cells were cultured for 3 d in the presence of BMP-2 (100 ng/mL) with the indicated concentration of OBWE. Osteoblast differentiation was visualized by alkaline phosphatase (ALP) staining. (**B**) ALP activity was monitored by measuring absorbance at 405 nm. ^###^
*p* < 0.001 (versus control); *** *p* < 0.001 (versus BMP-2–treated group). (**C**) Effects of OBWE on the viability of C2C12 cells were evaluated using the CCK-8 assay. Data are expressed as mean ± SD and are representative of at least three experiments.

**Figure 4 molecules-23-03119-f004:**
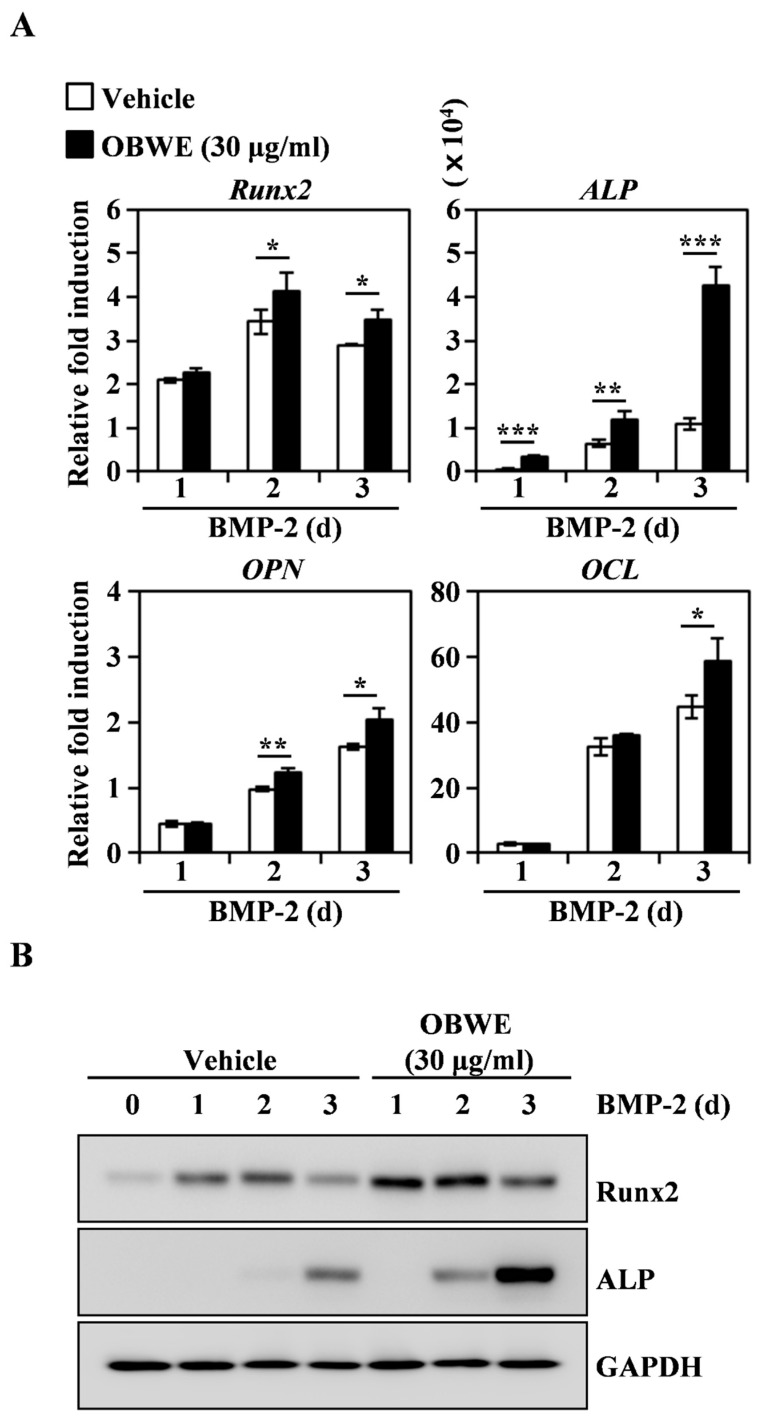
OBWE stimulates BMP-2–induced expression of Runx2. (**A**) C2C12 cells were stimulated in the presence of BMP-2 (100 ng/mL) with vehicle (water) or OBWE (30 µg/mL) for the indicated times. The mRNA expression levels were assessed using real-time PCR. *GAPDH* was used as the internal control. * *p* < 0.05; ** *p* < 0.01; *** *p* < 0.001 (versus vehicle control). (**B**) Effects of OBWE on the levels of Runx2 and ALP were evaluated by immune blot analysis. GAPDH was used as the internal control. One representative result from three independent experiments yielding similar results is shown.

**Figure 5 molecules-23-03119-f005:**
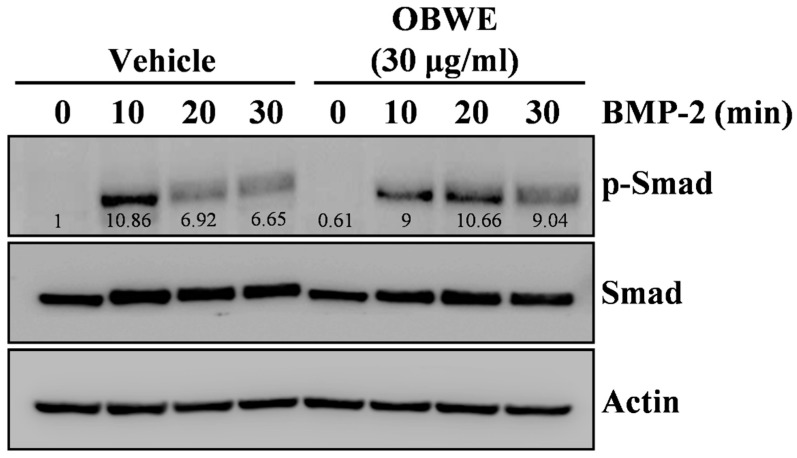
OBWE induces BMP-2–mediated phosphorylation of Smad signaling molecules. Following serum starvation for 1 d, C2C12 cells were pretreated with vehicle or OBWE (30 µg/mL) for 1 h prior to BMP-2 stimulation (100 ng/mL) for the indicated times. The expression levels of the signaling molecules were evaluated by Western blotting. Actin was used as the internal control.

**Figure 6 molecules-23-03119-f006:**
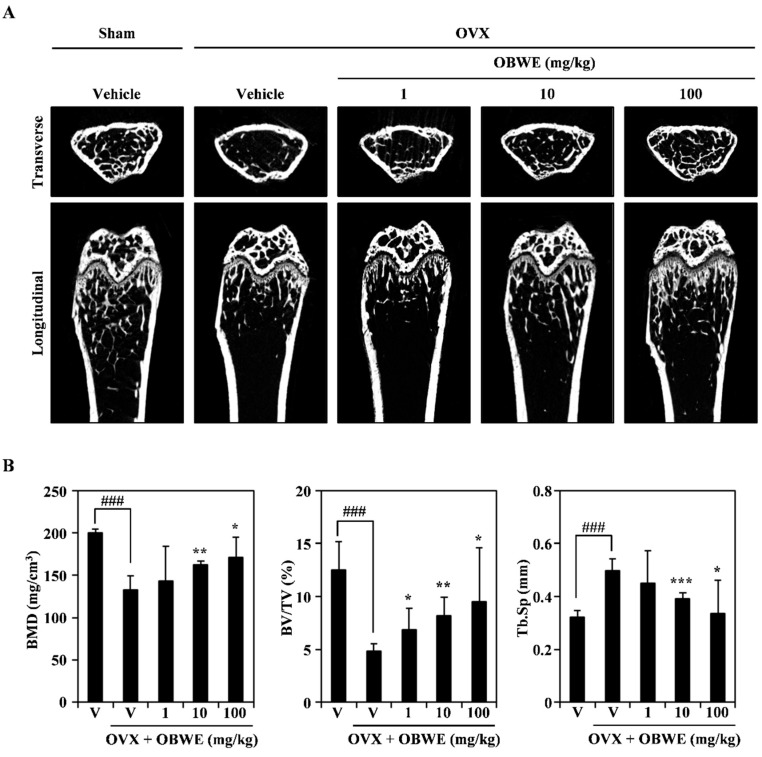
OBWE prevents OVX-induced bone loss in vivo. The effect of OBWE on OVX-induced bone loss was investigated in mice. (**A**) Micro-CT analysis-based images show transverse and longitudinal images of sham and OVX bone at 1 mg, 10 mg, and 100 mg/kg from left to right. (**B**) Bone mineral density (BMD), percent bone volume ratio (BV/TV), and trabecular separation (Tb. Sp) of femur measurements by 3D images analyzer. ^###^
*p* < 0.001 (versus the sham group); * *p* < 0.05; ** *p* < 0.01 (versus the OVX group).

**Table 1 molecules-23-03119-t001:** The primer sequences used in this study.

Target Gene	Forward Primer (5′–3′)	Reverse Primer (5′–3′)
*c-Fos*	CCAGTCAAGAGCATCAGCAA	AAGTAGTGCAGCCCGGAGTA
*NFATc1*	GGGTCAGTGTGACCGAAGAT	GGAAGTCAGAAGTGGGTGGA
*TRAP*	GATGACTTTGCCAGTCAGCA	ACATAGCCCACACCGTTCTC
*OSCAR*	AGGGAAACCTCATCCGTTTG	GAGCCGGAAATAAGGCACAG
*DC-STAMP*	CCAAGGAGTCGTCCATGATT	GGCTGCTTTGATCGTTTCTC
*Cathepsin K*	GGCCAACTCAAGAAGAAAAC	GTGCTTGCTTCCCTTCTGG
*Runx2*	GACTGTGGTTACCGTCATGGC	ACTTGGTTTTTCATAACAGCGGA
*ALP*	GATGGCGTATGCCTCCTGCA	CGGTGGTGGGCCACAAAAGG
*OPN*	ACACTTTCACTCCAATCGTCC	TGCCCTTTCCGTTGTTGTCC
*OCL*	AGGGAAACCTCATCCGTTG	GAGCCGGAAATAAGGCACAG
*GAPDH*	ACCACAGTCCATGCCATCAC	TCCACCACCCTGTTGCTGTA
*HPRT1*	TGCTCGAGATGTCATGAAGG	AGAGGTCCTTTTCACCAGCA

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
