# Peer review of "The Dual Role of Oat Bran Water Extract in Bone Homeostasis Through the Regulation of Osteoclastogenesis and Osteoblast Differentiation"

_molecules, 2018, doi:10.3390/molecules23123119_

Round 1

Reviewer 1 Report

I am pleased to review the assigned manuscript. Overall, a proposed review looks very good to me – Authors really did a wonderful job and presented very nice & relevant literature. The English is generally satisfactory, although there are some places where corrections and/or changes are required. The study does have few minor issues. The authors should improve the paper following these suggestions.

·         Abstract need bit of attention and should cover theme of whole manuscript

·         The introduction should be better organized. Some of the sentences are not well structured, should be clarified and rewritten. It is advice to link the story in a better way in an introduction to convey a proper message to readers.

·         Please add few suggested and latest references in background section

-       Immunity: Plants as Effective Mediators.

-       Natural polyphenols: An overview

-       Ethnic and paleolithic diet: Where do they stand in inflammation alleviation? A discussion.

·         Result and discussion is good, it’s well written and well organized but you can focus only latest reference in discussion if you want.

·         Please recheck the reference style, some of the references are not according to the journal instructions.

·         In general, manuscript looks good to me and authors did a wonderful job. I am happy to review the final version before acceptance.

Author Response

Dear reviewer,

We appreciate the thoughtful comments. Please find our response to each comment below. Changes were yellow-highlighted in the revised manuscript.

1) Abstract need bit of attention and should cover theme of whole manuscript.

2) The introduction should be better organized. Some of the sentences are not well structured, should be clarified and rewritten. It is advice to link the story in a better way in an introduction to convey a proper message to readers.

3) Please add few suggested and latest references in background section.

4) Result and discussion is good, it’s well written and well organized but you can focus only latest reference in discussion if you want

A1~4) As recommended, we reflected your mention in the manuscripts.

5) Please recheck the reference style, some of the references are not according to the journal instructions.

A5) As recommended, we corrected part in the reference.

Reviewer 2 Report

Dear authors,

I appreciated having the opportunity to review your manuscript. 

This is an elegant and well-written manuscript presenting new data about the potential applications for the oat bran cereal derivates through a mechanistic approach and an in vivo mice model of postmenopausal osteoporosis with ovariectomy.  

I have a few minor suggestions and questions to improve the readability of the manuscript further:

It would be better not to say, at the end of the abstract (last phrase, lines 28 - 29), that the findings of this study have the potential to “combat” bone metabolic disorders. It seems to me that this statement needs to be more softened.

In lines 39 - 41. “The imbalance between bone formation and resorption leads to…”. I wouldn’t say such affirmation since RA, Paget´s disease and periodontal lesions are provoked by different pathophysiological mechanisms that through different forms affect bone resorption and formation but this may not be the leading cause.

I would suggest you      clarify in lines 221 and also in the description of Figure 6 (lines 227 - 232)      that animals received for 6 weeks, after day 1 of      surgery, intragastric injections of OBWE or vehicle, and not only in      the “Materials and Methods” section (as it is found in line 339).

I have a question regarding the preparation of the OBWE. Could you, please, explain if the extract with prethanol was combined with the water extract? (lines 238-241). Does the prethanol extraction was orally administered to mice?

Author Response

Dear reviewer,

We appreciate the thoughtful comments. Please find our response to each comment below. Changes were yellow-highlighted in the revised manuscript.

1) It would be better not to say, at the end of the abstract (last phrase, lines 28 - 29), that the findings of this study have the potential to “combat” bone metabolic disorders. It seems to me that this statement needs to be more softened.

A1) As recommended, we reflected your mention in the abstract.

2) In lines 39 - 41. “The imbalance between bone formation and resorption leads to…”. I wouldn’t say such affirmation since RA, Paget´s disease and periodontal lesions are provoked by different pathophysiological mechanisms that through different forms affect bone resorption and formation but this may not be the leading cause.

A2) As recommended, we reflected your mention in the introduction.

3) I would suggest you clarify in lines 221 and also in the description of Figure 6 (lines 227 - 232) that animals received for 6 weeks, after day 1 of surgery, intragastric injections of OBWE or vehicle, and not only in the “Materials and Methods” section (as it is found in line 339).

A3) As recommended, we reflected your mention in the manuscript.

4) I have a question regarding the preparation of the OBWE. Could you, please, explain if the extract with prethanol was combined with the water extract? (lines 238-241). Does the prethanol extraction was orally administered to mice?

A4) Prethanol and water extracts were not combined. As shown in the experimental method, the samples used in this study were extracted from prethanol extraction with water.

In the animal experiments, samples were administered by orally administration.